# Anterior and Posterior Approaches for 4-Level Degenerative Cervical Myelopathy: Low-Profile Cage Versus Cervical Pedicle Screws Fixation

**DOI:** 10.3390/jcm12020564

**Published:** 2023-01-10

**Authors:** Peng Zou, Rui Zhang, Jun-Song Yang, Er-Liang Li, Qian Zhang, Yuan-Ting Zhao

**Affiliations:** 1Department of Spine Surgery, Honghui Hospital, Xi’an Jiaotong University, Xi’an 710000, China; 2Department of Orthopaedics and Traumatology, School of Clinical Medicine, Li Ka Shing Faculty of Medicine, the University of Hong Kong, Hong Kong SAR, China

**Keywords:** degenerative cervical myelopathy, anterior cervical decompression and fusion, cervical pedicle screws fixation

## Abstract

(1) Background: The choice of surgical access for 4-level degenerative cervical myelopathy (DCM) remains controversial, and the clinical and radiological outcomes of anterior surgery using a low-profile cage (Low-P) versus posterior surgery using cervical pedicle screw fixation (CPS) have not been compared. (2) Methods: This is a retrospective controlled study conducted between January 2019 and June 2021 of 72 patients with 4-level DCM who underwent ACDF using a low-profile cage (*n* = 39) or laminectomy and instrument fusion using CPS (*n* = 33). The minimum follow-up time was 12 months. The outcomes were C2–7Cobb angle, C2–7sagittal vertical axis (SVA) fusion rate, the Japanese Orthopedic Association (JOA) score, pain visual analog scale (VAS), neck disability index (NDI), and complications. (3) Results: Both anterior and posterior procedures significantly improved the patients’ quality-of-life parameters. Anterior cervical convexity and SVA significantly increased in both groups, but the SVA was greater in the posterior group than in the anterior group (*p* < 0.001). The C2–7 Cobb angle significantly improved in both groups postoperatively, and at the final follow-up, there was a slight but nonsignificant reduction in cervical lordosis in both groups (*p* = 0.567). There was a longer operative time, less intraoperative blood loss, and reduced mean hospital stay in the anterior group compared to the posterior group, with two cases of postoperative hematoma requiring a second operation, two cases of axial pain (AP), five cases of dysphagia, two cases of c5 palsy in the anterior group, and four cases of axial pain, and three cases of c5 palsy in the posterior group. According to Bridwell fusion grade, anterior fusion reached grade I in 28 cases (71.8%) and grade II in 10 cases (25.6%) in the anterior group, and posterior fusion reached grade I in 25 cases (75.8%) and grade II in 8 cases (24.2%) in the posterior group. (4) Conclusions: There was no difference between the anterior and posterior surgical approaches for MDCM in terms of improvement in neurological function. Posterior surgery using CPS achieved similar recovery of cervical anterior convexity as anterior surgery with a shorter operative time but was more invasive and had a greater increase in SVA. The use of Low-P in anterior surgery reduced the incidence of dysphagia and cage subsidence and was less invasive, but with a longer operative time.

## 1. Introduction

The incidence of degenerative cervical myelopathy (DCM) is increasing with the aging population, and it is well-recognized as a primary cause of spinal cord dysfunction. Surgical decompression can enhance neurological function and prevent future deterioration [1,2,3]. Although surgical treatments have greatly enhanced over the last few decades, there is still no consensus approach for treating multilevel DCM (MDCM), particularly 4-level DCM [3,4,5,6].

Treatment of MDCM with ACDF using a conventional plate-cage structure is associated with improved neurological function and better kyphosis correction. However, a higher number of surgical levels, high-profile internal fixation, and prolonged muscle traction may lead to a higher rate of cage subsidence and nonunion, as well as a higher incidence of dysphagia [4,5,7]. Zero-profile cages have been trialed in MDCM, which help reduce the risk of developing dysphagia, but higher-risk cage subsidence rates and cervical alignment problems must be considered [8,9,10]. Posterior cervical laminectomy and instrumented fusion (PCLF) can achieve indirect decompression by causing the spinal cord to drift posteriorly and is mostly used in patients with MDCM [1,3,11], but the widely used traditional lateral mass screw fixation is not very stable and does not provide satisfactory correction of kyphosis [12,13,14]. 

We hypothesized that the use of low-profile cage (Low-P) in anterior surgery for 4-level DCM would reduce the incidence of postoperative dysphagia and cage subsidence; and that the use of cervical pedicle screw fixation (CPS) in posterior surgery for 4-level DCM would provide satisfactory correction of kyphosis. Therefore, we retrospectively compared the clinical and imaging outcomes of Low-P and CPS in 4-level DCM. 

## 2. Materials and Methods

A total of 74 patients with grade 4 DCM who underwent surgery at our institution between January 2019 and June 2021 were enrolled in this study, with 2 patients lost to follow-up, so 72 patients completed the trial.

The diagnosis was based on radiological (X-ray, CT, MRI), physical examinations, and medical history. Inclusion criteria were as follows: positive signs of spinal cord compression consistent with radiological and ineffective conservative treatment; 4-level DCM of C3−7; ≤50% canal encroachment; soft kyphosis (hyperextension X-ray showing >50% reduction in kyphotic angle); follow-up for at least 12 months. The exclusion criteria were: patients with severe osteoporosis, developmental cervical spinal stenosis, continuous post longitudinal ligament ossification, yellow ligament hypertrophy, combined diabetes, cerebral infarction, peripheral neuritis, motor neuron, and other medical diseases resulting in poor postoperative results or inability to determine the efficacy, spinal cord tumor, spinal cord cavity, and congenital malformation, and rigid cervical kyphosis.

### 2.1. Surgical Techniques

Anterior Group: the anterior approach surgery was performed according to Smith et al. [15]. All patients underwent general anesthesia and neurophysiological monitoring and were placed in a supine position with the neck slightly posteriorly extended. An oblique incision was made at the internal border of the sternocleidomastoid muscle to expose the target segment. The intervertebral space with the most significant compression was treated first, and the intervertebral space was propped open to completely remove the degenerated disc tissue, cartilaginous endplates, and hyperplastic bone, preserving the bony endplates. The posterior edge of the vertebral body was carefully abraded with a high-speed burr, and the posterior longitudinal ligament was severed and adequately decompressed to the bilateral Luschka joint in patients with comorbid radicular symptoms. A low-profile cage (carmen Integral Cervical Interbody Fusion System, Sanyou, Shanghai, China) (Figure 1) was implanted with artificial bone and some autologous bone, then inserted into the decompressed intervertebral space. Other involved levels were treated using the same method.

Posterior Group: the posterior approach surgery was performed according to Abumi [16]. All patients required preoperative cervical computed tomography (CT) and CT angiography (CTA) to exclude patients with incomplete pedicles, pedicles less than 4 mm diameter, and vertebral artery variants. Patients were anesthetized in the prone position maintaining the cervical spine in a neutral position, and the head was immobilized using a U-shaped pillow. The cervical spine sequence was fluoroscopically visualized using the C-arm, and adjustments were made based on the fluoroscopic findings. The insertion holes for the pedicle screws in C3–6 were selected at the intersection of the vertical line 2 to 3 mm below the superior articular eminence and the external 1/4 of the superior articular eminence; the screw angle was 40° to 45° in the sagittal plane, and the horizontal plane was parallel to the endplate; the insertion holes in C7 were selected at the intersection of the vertical line of the superior articular eminence and 2 to 3 mm below the superior articular eminence; the screw angle was 30° to 40° in the sagittal plane, and the horizontal plane was parallel to the endplate. A probe was used to detect cortical breakdown of the pedicle wall before screw insertion. Hand tapping rather than drilling was used throughout the process of creating the insertion hole. After placement of the screw, the lamina was removed using an ultrasonic bone knife, and the width of the removed lamina was approximately 15 mm. After spinal cord decompression, the rods were bent to form a curve and installed at the end of the screw, with the autologous bone fragments or allograft bone laid flat in the joint space and the tail of the screw.

### 2.2. Postoperative Management

Postoperative anti-inflammatory and nerve nutrition were administered to patients to prevent spinal nerve edema. The patients were encouraged to gradually get out of bed and resume activity 2–3 days after surgery under the protection of cervical braces. All patients were required to wear a cervical brace for 4–6 weeks.

### 2.3. Clinical Evaluation

(1) Surgical condition: the operation time and bleeding volume of patients in both groups, and the incidence of intraoperative and postoperative related complications (incisional hematoma, dysphagia, hoarseness, cerebrospinal fluid leakage, wound infection, aggravation of spinal nerve injury, AP, C5 palsy, etc.) were recorded. (2) Radiological evaluation: the C2–7 Cobb angle of the cervical spine and C2–7 sagittal vertical axis (SVA) were measured preoperatively, postoperatively, and at the last follow-up. The C2–7 Cobb angle [17] is the angle between the vertical line connecting the C2 upper endplate and the C7 lower endplate measured on a standard lateral X-ray, and the angle was positive for cervical lordosis and negative for cervical kyphosis. The C2–7SVA [18] is the distance between the vertical line through the center of C2 and the vertical line through the upper corner after C7. (3) Clinical functional evaluation: the visual analog scale (VAS) was used to score neck, shoulder, and upper limb pain before and after surgery and at the last follow-up. The incidence of axial pain was recorded at the follow-up and cervical axial pain (AP) [19] was defined as prolonged postoperative pain and stiffness in the neck and back of the shoulder with soreness, heaviness, and muscle spasm. The JOA score was used to evaluate the neurological efficacy and the recovery rate (RR) of the JOA score = (postoperative score − preoperative score)/(17 − preoperative score) × 100%.

### 2.4. Statistical Analysis

R version 3.5.3 (R Development Core Team, Auckland, New Zealand) was used for statistical analysis. An analysis of variance was used to compare the means of continuous variables with normal distributions. In cases of a normal distribution, continuous variables were summarized by mean ± standard deviation. Student *t*-tests were used to compare independent and paired *t*-tests for matched samples. If not normally distributed, the median (interquartile range) was used, and the Mann–Whitney U test was applied. Categorical data were expressed as numbers and percentages and assessed by the chi-square test. The nonparametric test Ridit analysis was used to compare differences in rank variables.

## 3. Results

### 3.1. Characteristics of the Study Participants

All 72 patients completed at least 12 months of follow-up. Table 1 summarizes the demographic parameters of the two groups, without statistical differences in the main baseline parameters.

### 3.2. Clinical Outcomes

The anterior group had a significantly longer mean operative time (126.5 ± 19.1 min vs. 110.3 ± 7.7 min), less mean blood loss (*p* < 0.001), and shorter hospital stay (*p* < 0.001) compared to the posterior group (Table 2). At the final follow-up, the VAS of neck pain, NDI score, and JOA score were significantly improved in both groups postoperatively (Table 2).

### 3.3. Radiographic Evaluation

The average postoperative C2–7 cobb angle improved from 4.1° to −12.3° in the anterior group and from 3.8° to −12° in the posterior group, without significant differences between the groups (Table 3). At the final follow-up, the mean SVA increased from 14.8 mm to 19.0 mm in the anterior group and from 15.6 mm to 21.8 mm in the posterior group and was significantly different between the two groups (Table 3). Figure 2 shows the recovery of the patient’s cobb angle and the increase in SVA after the anterior and posterior surgeries.

### 3.4. Complications

There were two cases of postoperative hematoma requiring secondary surgery, five cases of dysphagia, two cases of axial pain, and two cases of c5 palsy in the anterior group, with four cases of axial pain and three cases of C5 palsy in the posterior group. According to the Bridwell fusion grade [20], anterior fusion reached grade I in 28 cases (71.8%) and grade II in 10 cases (25.6%) in the anterior group, and posterior fusion reached grade I in 25 cases (75.8%) and grade II in 8 cases (24.2%) in the posterior group. The incidence of dysphagia was significantly lower (*p* = 0.033) in the posterior group than in the anterior group (Table 4).

## 4. Discussion

### 4.1. Correcting Sagittal Plane Deformities

The main goals of surgical treatment of DCM should be to adequately decompress, correct the deformity and re-establish stability [21,22,23,24,25]. Anterior surgery and CPS effectively reconstruct and maintain cervical lordosis, and our results are consistent with the literature [12,14,16,17,26,27]. In the present study, the C2–7 cobb angle was significantly increased in both groups, without significant loss of cervical lordosis in either group at the final follow-up. 

In the posterior procedure, the CPS can fix the anterior and posterior columns of the cervical spine, which has good biomechanical stability and greater pullout strength, and the reconstructed cervical lordosis allows the weight-bearing axis to revert to the reconstructed posterior structure [17]. In the anterior procedure, the Low-P requires multiple independent cages for continuous fixation during surgery for MDCM treatment. Each adjacent stage is secured by three screws, which allows for a better fit between the plate and the vertebral body and between the cage and the upper and lower endplates. This, combined with the 4° lordotic angle of each cage, increases cervical lordosis.

Cervical C2–7 SVA is often used to measure cervical sagittal alignment [24], and Tang [13] and Hyun et al. [28] reported that cervical C2–7 SVA > 40 mm and 43.5 mm, respectively, were associated with poorer NDI scores in posterior multistage fusion surgery. Yang [29], Jeon [30], and Kwon et al. [31] concluded that multilevel ACDF has insignificant effects on postoperative cervical alignment relative to posterior surgery, similar to our study. Compared with the preoperative period, SVA was significantly increased in both the anterior and posterior groups, but the postoperative SVA was significantly greater in the posterior group than in the anterior group, without difference in the NDI and JOA scores between the two groups. The supine position for ACDF is easier to obtain a cervical alignment close to physiological cervical lordosis, whereas posterior surgery must be performed in a prone position without good postural positioning and fixation, which may affect the alignment of the cervical spine [12,30].

### 4.2. Neurological Improvement and Surgical Features

In this study, the anterior group had a longer operative time than the posterior group, but significantly less operative bleeding and mean length of stay. This is related to the characteristics of the two procedures, with the anterior approach causing less damage to muscle tissue and less intraoperative bleeding, whereas more discs need to be removed for decompression, which results in a longer operative time during the posterior procedure. In terms of parameters related to neurological recovery, JOA, NDI, and VAS revealed good results in both groups compared to the preoperative period, without significant difference between the two groups. This indicates no difference in postoperative neurological improvement between the two groups, which is consistent with the findings of Fehlings et al., Li et al., and Luo et al. [32,33,34].

### 4.3. Complications

Postoperative AP is highly prevalent in posterior cervical spine surgery, with a reported incidence of 5.2–61.5% [35,36]. In our study, four (4/33, 12.1%) cases of postoperative AP were found in the posterior group and two (2/39, 5.1%) cases occurred in the anterior group. The AP complication rate was lower than that reported in the literature, possibly because we avoided excessive use of the electric knife when performing the posterior approach and tried to preserve the C7 spinous process and the cervical extensor attachment point and reconstructed the semispinalis cervicis on the C2 spinous process. This preserves the C2 and C7 spinous processes as well as the muscle attachment points, which can alleviate postoperative cervical muscle strength imbalances to some extent and reduce the risk of AP [37]. There were two cases of AP in the anterior group, possibly related to the over-propping of the vertebral space. With multiple cage placements, the plate requires more space, and if the patient has a small cervical vertebral body, there is a risk of over-propping the intervertebral space. Excessive curvature has the potential to cause small joint distractions or muscle spasms, which may lead to AP [19].

The risk of postoperative dysphagia, cage subsidence, and nonunion may be higher with multi-stage ACDF [4,5,7]. In our study, no cage subsidence was found in the anterior group, and according to Bridwell fusion grade [20], anterior fusion reached grade I in 28 cases (71.8%) and grade II in 10 cases (25.6%) in the anterior group, and posterior fusion reached grade I in 25 cases (75.8%) and grade II in 8 cases (24.2%) in the posterior group. Compared to the zero-profile and plate-cage, the low profile preserves the anterior panel to improve cervical stability, reduce subsidence, and improve the fusion rate. It also reduces the profile of the anterior panel, thereby reducing mechanical irritation to the esophagus and reducing the risk of postoperative dysphagia [38]. Five patients (12.8%) who underwent the anterior approach had transient dysphagia one week postoperatively but recovered after subsequent treatment. Based on the incidence of 904 patients in the meta-analysis of Guo et al., our incidence was similar to the 13.97% of zero-p and lower than the 26.01% of plate-cage [10]. This suggests that Low-P can achieve a similar effect to zero-p in reducing the incidence of postoperative dysphagia. There were no cases of dysphagia in the posterior approach group, two cases of C5 palsy in the anterior approach group, and three cases in the posterior approach group, without significant differences between groups.

This study has several limitations. First, it was a single-center retrospective study, and the small sample size and selection bias may affect the conclusions; second, the cases did not include patients with rigid kyphosis; Finally, multiple placements of Low-P in the anterior cervical require some space, and patients with small vertebrae may have difficulty with placement. Therefore, a large, multicenter, prospective study in future studies is warranted.

## 5. Conclusions

There was no difference between the anterior and posterior surgical approaches for MDCM in terms of improvement in neurological function. Posterior surgery using CPS achieved similar recovery of cervical anterior convexity as anterior surgery with a shorter operative time but was more invasive and had a greater increase in SVA. The use of Low-P in anterior surgery reduced the incidence of dysphagia and cage subsidence and was less invasive, but with a longer operative time.

## Figures and Tables

**Figure 1 jcm-12-00564-f001:**
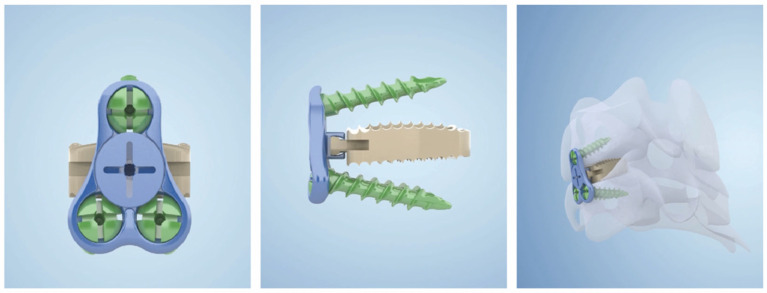
Low-profile cage (Shanghai Sanyou, carmen Integral Cervical Interbody Fusion System).

**Figure 2 jcm-12-00564-f002:**
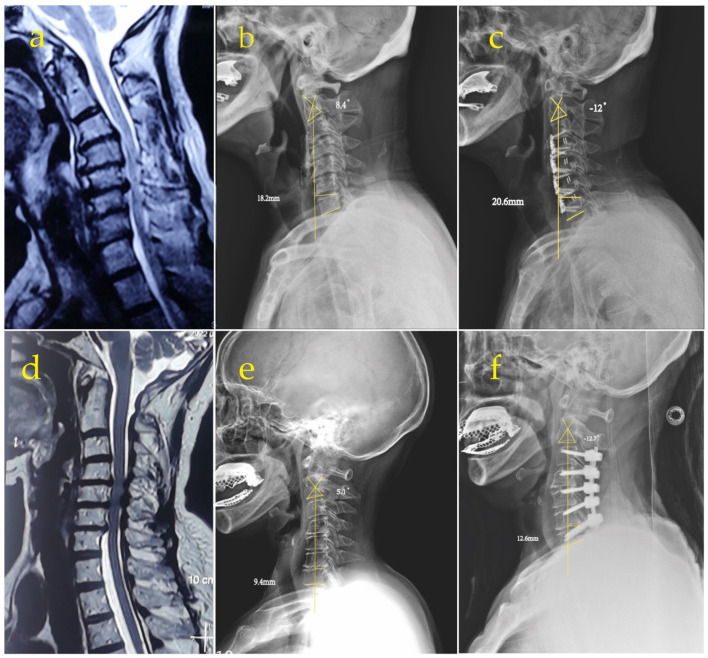
(**a**) Preoperative MRI showing cervical spinal stenosis and myelopathy. The preoperative (**b**) and postoperative (**c**) plain films displayed that the anterior convexity of C2–7 was restored after anterior surgery and the SVA increased compared to preoperatively. (**d**) Preoperative MRI showing cervical spinal stenosis and myelopathy. The preoperative (**e**) and postoperative (**f**) plain films revealed that the anterior convexity of C2–7 was restored after posterior surgery and the SVA increased compared to preoperatively.

**Table 1 jcm-12-00564-t001:** Patient demographic data.

Characteristic	Anterior Group (*n* = 39)	Posterior Group (*n* = 33)	*p*-Value
Sex (male/female)	17/22	18/15	0.354
Age (years)	69.1 ± 5.3	68.4 ± 5.4	0.563
Disease duration	24.8 ± 3.9	25.9 ± 4.2	0.266
Follow-up time (months)	18.7 ± 4.3	19.8 ± 4.0	0.300

Data are expressed as mean ± SD.

**Table 2 jcm-12-00564-t002:** Functional outcomes of both groups.

Variable	Anterior Group (*n* = 39)	Posterior Group (*n* = 33)	*p*-Value
Operation time (min)	126.5 ± 19.1	110.3 ± 7.7	<0.001
Blood loss (mL)	213.0 ± 90.7	286.3 ± 63.0	<0.001
postoperative hospital stays	5.1 ± 1.0	6.3 ± 1.3	<0.001
VAS			
Preoperative	4.2 ± 0.7	4.1 ± 0.8	0.488
Postoperative	2.0 ± 0.4 ^※^	1.9 ± 0.7 ^※^	0.244
Follow-up	1.5 ± 0.4 ^‡^	1.4 ± 0.4 ^‡^	0.341
NDI			
Preoperative	20.2 ± 1.9	19.4 ± 3.7	0.231
Follow-up	5.0 ± 1.1 ^※^	5.2 ± 1.2 ^※^	0.428
JOA scale score			
Preoperative	9.8 ± 0.9	10.0 ± 0.9	0.418
Follow-up	13.6 ± 0.8 ^※^	13.7 ± 0.9 ^※^	0.523
RR	52.4 ± 10.7	53.4 ± 12.1	0.725

Data are expressed as mean ± S.D.; VAS, visual analog scale; J.O.A., Japanese Orthopedic Association; NDI, neck disability index; RR, recovery rate. ^※^ *p* < 0.05, comparison of preoperative and postoperative variables within groups. ^‡^ *p* < 0.05, comparison of postoperative and final follow-up variables between groups.

**Table 3 jcm-12-00564-t003:** Radiologic outcomes of both groups.

Variable	Anterior Group (*n* = 39)	Posterior Group (*n* = 33)	*p*-Value
C2–7 Cobb angle			
Preoperative	4.1 ± 5.9	3.8 ± 6.1	0.864
Postoperative	−12.3 ± 2.6 ^※^	−12.0 ± 2.0 ^※^	0.500
Follow-up	−11.5 ± 2.2 ^†^	−11.2 ± 1.9 ^†^	0.567
SVA (mm)			
Preoperative	14.8 ± 2.7	15.6 ± 3.4	0.287
Postoperative	18.1 ± 2.9 ^※^	20.6 ± 2.5 ^※^	<0.001
Follow-up	19.0 ± 2.7 ^†^	21.8 ± 2.4 ^†^	<0.001

^※^ *p* < 0.05, comparison of preoperative and postoperative variables within groups. ^†^ *p* > 0.05, comparison of postoperative and final follow-up variables between groups.

**Table 4 jcm-12-00564-t004:** Complications occurred in the two groups.

Variable	Anterior Group (*n* = 39)	Posterior Group (*n* = 33)	*p*-Value
Dysphagia	5	0	0.033
C5 palsy	2	3	0.510
Axial pain	2	4	0.285
CSF leakage	5	3	0.616
Wound infection	0	1	0.274
postoperative hematoma	2	0	0.187
Total	16	11	0.502
Bridwell fusion grade (%) *			0.737
I	28 (71.8%)	25 (75.8%)	N/A
II	10 (25.6%)	8 (24.2%)	N/A
III	1 (2.5%)	0	N/A
IV	0	0	N/A

CSF, Cerebrospinal fluid; N/A, not applicable. *, comparison of fusion levels variables between groups using Ridit analysis.

## Data Availability

The datasets generated during the current study are publicly available by email to doczhaoyuanting@163.com.

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
