# Peer review of "Anterior and Posterior Approaches for 4-Level Degenerative Cervical Myelopathy: Low-Profile Cage Versus Cervical Pedicle Screws Fixation"

_jcm, 2023, doi:10.3390/jcm12020564_

Round 1

Reviewer 1 Report

1. The abbreviation appears from the beginning without explanation. Even if it was explained in the abstract, if abbreviations appear for the first time in the manuscript body, please write it again. 

For example, low-p, CPS in line 58, SVA in line 121. 

Also you used an abbreviation in the front but not in the back. A low-profile in line 199. 

2. For table 2 and 3 , The p-value of 0.000 is somewhat imprecise. Change it to <0.001.

3. The description of Figure 2 is omitted from the manuscript body. I think it should express in 3.3. Radiograhic evaluation. 

4. I  think the research results are neat, but it's a bit strange that there are no limitations in the thesis.

Author Response

Many thanks to the reviewers for their valid suggestions.

We have revised the manuscript based on the reviewers' comments and marked it in red.

  1. The abbreviation appears from the beginning without explanation. Even if it was explained in the abstract, if abbreviations appear for the first time in the manuscript body, please write it again.

For example, low-p, CPS in line 58, SVA in line 121.

Also you used an abbreviation in the front but not in the back. A low-profile in line 199.

ANSWER: A correction has been made in the manuscript.

  1. For table 2 and 3 , The p-value of 0.000 is somewhat imprecise. Change it to <0.001.

ANSWER: A correction has been made in the table.

  1. The description of Figure 2 is omitted from the manuscript body. I think it should express in 3.3. Radiograhic evaluation.

ANSWER: Relevant content has been added in 3.3.

  1. I think the research results are neat, but it's a bit strange that there are no limitations in the thesis.

ANSWER: limitations have been added to the discussion section.

Thanks again to the reviewers.

Reviewer 2 Report

The article is interesting but needs some corrections.

Please match the purpose of your research with the topic.

Please include your research questions and hypotheses in the Introduction.

Please provide the number and date of the bioethics committee.

In the discussion, please indicate the weaknesses of the research.

Conclusions should answer the research questions.

Author Response

Many thanks to the reviewers for their valid suggestions.

We have revised the manuscript by the reviewers' suggestions and have marked it in red.

Please match the purpose of your research with the topic.

Please include your research questions and hypotheses in the Introduction.

Please provide the number and date of the bioethics committee.

In the discussion, please indicate the weaknesses of the research.

Conclusions should answer the research questions.

A: Corrections were made in the manuscript on the purpose of the study, research questions and hypotheses, study limitations, and conclusions.

The bioethics committee number and date are mentioned in the statement at the end of the manuscript. Institutional Review Board Statement: The study was conducted following the Declaration of Helsinki and approved by the Institutional Review Board (or Ethics Committee) of Xi’an Honghui Hospital (2019 Ethics Approval No. 011)

Thanks again to the reviewers.

Reviewer 3 Report

This is a well structured article. The comparison between posterior vs anterior approaches is indeed a very hot topic. The decision to go posterior or anterior would depend on whether the main problem is, i.e anterior compression or posterior elements (yellow ligament) responsible for that compression. That is, of course, a variable a retrospective study can't control. Nevertheless, is interesting to know that they are very similar in outcomes (pain, disability, etc) while having their own particular complications. Regarding the number of levels, it is true that complications are higher, but this is not necessarily worse than going posterior. The thing here is that the authors used a low-profile multilevel cage without using anterior cervical plates, which can contribute to these results when compared to, for example, cages and anterior cervical plates. 

Author Response

We thank the reviewers for their recognition of the significance of our research, which will always inspire us for future research. Once again, we thank the reviewers for their valid suggestions.

Round 2

Reviewer 2 Report

I accept the revised article